# Some Considerations on Learning to Explore via Meta-Reinforcement Learning

**Bradly C. Stadie**[*]
UC Berkeley

**Ge Yang**[*]
University of Chicago

**Rein Houthooft**
OpenAI

**Xi Chen**
Covariant.ai

**Yan Duan**
Covariant.ai

**Yuhuai Wu**
University of Toronto

**Pieter Abbeel**
UC Berkeley

**Ilya Sutskever**
OpenAI

## Abstract

We interpret meta-reinforcement learning as the problem of learning how to quickly find a good sampling distribution in a new environment. This interpretation leads to the development of two new meta-reinforcement learning algorithms: E-MAML and E-RL$^2$. Results are presented on a new environment we call 'Krazy World': a difficult high-dimensional gridworld which is designed to highlight the importance of correctly differentiating through sampling distributions in meta-reinforcement learning. Further results are presented on a set of maze environments. We show E-MAML and E-RL$^2$ deliver better performance than baseline algorithms on both tasks.

## 1 Introduction

Reinforcement learning can be thought of as a procedure wherein an agent bias its sampling process towards areas with higher rewards. This sampling process is embodied as the policy $\pi$, which is responsible for outputting an action $a$ conditioned on past environmental states $\{s\}$. Such action affects changes in the distribution of the next state $s' \sim T(s, a)$. As a result, it is natural to identify the policy $\pi$ with a sampling distribution over the state space.

This perspective highlights a key difference between reinforcement learning and supervised learning: In supervised learning, the data is sampled from a fixed set. The i.i.d. assumption implies that the model does not affect the underlying distribution. In reinforcement learning, however, the very goal is to learn a policy $\pi(a|s)$ that can manipulate the sampled states $P_\pi(s)$ to the agent's advantage.

This property of RL algorithms to affect their own data distribution during the learning process is particularly salient in the field of meta-reinforcement learning. Meta RL goes by many different names: learning to learn, multi-task learning, lifelong learning, transfer learning, etc [25, 26, 22, 21, 23, 12, 24, 39, 37]. The goal, however, is usually the same–we wish to train the agents to learn transferable knowledge that helps it generalize to new situations. The most straightforward way to tackle this problem is to explicitly optimize the agent to deliver good performance after some adaptation step. Typically, this adaptation step will take the agent's prior and use it to update its current policy to fit its current environment.

This problem definition induces an interesting consequence: during meta-learning, we are no longer under the obligation to optimize for maximal reward during training. Instead, we can optimize for a

---

[*]equal contribution, correspondence to {bstadie, ge.yang}@berkeley.edu
Code for Krazy World available at: https://github.com/bstadie/krazyworld
Code for meta RL algorithms available at: https://github.com/episodeyang/e-maml

sampling process that maximally informs the meta-learner how it should adapt to new environments. In the context of gradient based algorithms, this means that one principled approach for learning an optimal sampling strategy is to differentiate the meta RL agent's per-task sampling process with respect to the goal of maximizing the reward attained by the agent post-adaptation. To the best of our knowledge, such a scheme is hitherto unexplored.

In this paper, we derive an algorithm for gradient-based meta-learning that explicitly optimizes the per-task sampling distributions during adaptation with respect to the expected future returns produced by it post-adaptation self. This algorithm is closely related to the recently proposed MAML algorithm [7]. For reasons that will become clear later, we call this algorithm E-MAML. Inspired by this algorithm, we develop a less principled extension of RL$^2$ that we call E-RL$^2$. To demonstrate that this method learns more transferable structures for meta adaptation, we propose a new, high-dimensional, and dynamically-changing set of tasks over a domain we call "Krazy-World". "Krazy-World" requires the agent to learn high-level structures, and is much more challenging for state-of-the-art RL algorithms than simple locomotion tasks.[2].

We show that our algorithms outperform baselines on "Krazy-World". To verify we are not over-fitting to this environment, we also present results on a set of maze environments.

## 2 Preliminaries

**Reinforcement Learning Notation:** Let $M = (\mathcal{S}, \mathcal{A}, \mathcal{P}, R, \rho_0, \gamma, T)$ represent a discrete-time finite-horizon discounted Markov decision process (MDP). The elements of $M$ have the following definitions: $\mathcal{S}$ is a state set, $\mathcal{A}$ an action set, $\mathcal{P} : \mathcal{S} \times \mathcal{A} \times \mathcal{S} \to \mathbb{R}_+$ a transition probability distribution, $R : \mathcal{S} \times \mathcal{A} \to \mathbb{R}$ a reward function, $\rho_0 : \mathcal{S} \to \mathbb{R}_+$ an initial state distribution, $\gamma \in [0, 1]$ a discount factor, and $T$ is the horizon. Occasionally we use a loss function $\mathcal{L}(s) = -R(s)$ rather than the reward $R$. In a classical reinforcement learning setting, we optimize to obtain a set of parameters $\theta$ which maximize the expected discounted return under the policy $\pi_\theta : \mathcal{S} \times \mathcal{A} \to \mathbb{R}_+$. That is, we optimize to obtain $\theta$ that maximizes $\eta(\pi_\theta) = \mathbb{E}_{\pi_\theta}[\sum_{t=0}^{T} \gamma^t r(s_t)]$, where $s_0 \sim \rho_0(s_0)$, $a_t \sim \pi_\theta(a_t|s_t)$, and $s_{t+1} \sim \mathcal{P}(s_{t+1}|s_t, a_t)$.

**The Meta Reinforcement Learning Objective:** In meta reinforcement learning, we consider a family of MDPs $\mathcal{M} = \{M_i\}_{i=1}^N$ which comprise a distribution of tasks. The goal of meta RL is to find a policy $\pi_\theta$ and paired update method $U$ such that, when $M_i \sim \mathcal{M}$ is sampled, $\pi_{U(\theta)}$ solves $M_i$ quickly. The word *quickly* is key: By quickly, we mean orders of magnitude more sample efficient than simply solving $M_i$ with policy gradient or value iteration methods from scratch. For example, in an environment where policy gradients require over 100,000 samples to produce good returns, an ideal meta RL algorithm should solve these tasks by collecting less than 10 trajectories. The assumption is that, if an algorithm can solve a problem with so few samples, then it might be 'learning to learn.' That is, the agent is not learning how to master a particular task but rather how to quickly adapt to new ones. The objective can be written cleanly as

$$\min_\theta \sum_{M_i} \mathbb{E}_{\pi_{U(\theta)}} [\mathcal{L}_{M_i}] \tag{1}$$

This objective is similar to the one that appears in MAML [7], which we will discuss further below. In MAML, $U$ is chosen to be the stochastic gradient descent operator parameterized by the task.

## 3 Problem Statement and Algorithms

### 3.1 Fixing the Sampling Problem with E-MAML

We can expand the expectation from (1) into the integral form

$$\int R(\tau)\pi_{U(\theta)}(\tau)\mathrm{d}\tau \tag{2}$$

It is true that the objective (1) can be optimized by taking a derivative of this integral with respect to $\theta$ and carrying out a standard REINFORCE style analysis to obtain a tractable expression for the

gradient [40]. However, this decision might be sub-optimal.

Our key insight is to recall the sampling process interpretation of RL. In this interpretation, the policy $\pi_\theta$ implicitly defines a sampling process over the state space. Under this interpretation, meta RL tries to learn a strategy for quickly generating good per-task sampling distributions. For this learning process to work, it needs to receive a signal from each per-task sampling distribution which measures its propensity to positively impact the meta-learning process. Such a term does not make an appearance when directly optimizing (1). Put more succinctly, **directly optimizing (1) will not account for the impact of the original sampling distribution $\pi_\theta$ on the future rewards** $R(\tau), \tau \sim \pi_{U(\theta,\bar{\tau})}$. Concretely, we would like to account for the fact that the samples $\bar{\tau}$ drawn under $\pi_\theta$ will impact the final returns $R(\tau)$ by influencing the initial update $U(\theta, \bar{\tau})$. Making this change will allow initial samples $\bar{\tau} \sim \pi_\theta$ to be reinforced by the expected future returns after the sampling update $R(\tau)$. Under this scheme, the initial samples $\bar{\tau}$ are encouraged to cover the state space enough to ensure that the update $U(\theta, \bar{\tau})$ is maximally effective.

Including this dependency can be done by writing the modified expectation as

$$\iint R(\tau)\pi_{U(\theta,\bar{\tau})}(\tau)\pi_\theta(\bar{\tau})\mathrm{d}\bar{\tau}\mathrm{d}\tau \tag{3}$$

This provides an expression for computing the gradient which correctly accounts for the dependence on the initial sampling distribution.

We now find ourselves wishing to find a tractable expression for the gradient of (3). This can be done quite smoothly by applying the product rule under the integral sign and going through the REINFORCE style derivation twice to arrive at a two term expression

$$\frac{\partial}{\partial \theta} \iint R(\tau)\pi_{U(\theta,\bar{\tau})}(\tau)\pi_\theta(\bar{\tau})\mathrm{d}\bar{\tau}\mathrm{d}\tau$$

$$= \iint R(\tau)\left[\pi_\theta(\bar{\tau})\frac{\partial}{\partial \theta}\pi_{U(\theta,\bar{\tau})}(\tau) + \pi_{U(\theta,\bar{\tau})}(\tau)\frac{\partial}{\partial \theta}\pi_\theta(\bar{\tau})\right]\mathrm{d}\bar{\tau}\mathrm{d}\tau$$

$$\approx \frac{1}{T}\sum_{i=1}^{T} R(\tau^i)\frac{\partial}{\partial \theta}\log \pi_{U(\theta,\bar{\tau})}(\tau^i) + \frac{1}{T}\sum_{i=1}^{T} R(\tau^i)\frac{\partial}{\partial \theta}\log \pi_\theta(\bar{\tau}^i) \left.\right|_{\substack{\bar{\tau}^i \sim \pi_\theta \\ \tau^i \sim \pi_{U(\theta,\bar{\tau})}}} \tag{4}$$

The term on the left is precisely the original MAML algorithm [7]. This term encourages the agent to take update steps $U$ that achieve good final rewards. The second term encourages the agent to take actions such that the eventual meta-update yields good rewards (crucially, it does not try and exploit the reward signal under its own trajectory $\bar{\tau}$). In our original derivation of this algorithm, we felt this term would afford the the policy the opportunity to be more exploratory, as it will attempt to deliver the maximal amount of information useful for the future rewards $R(\tau)$ without worrying about its own rewards $R(\bar{\tau})$. Since we felt this algorithm augments MAML by adding in an exploratory term, we called it E-MAML. At present, the validity of this interpretation remains an open question.

For the experiments presented in this paper, we will assume that the operator $U$ that is utilized in MAML and E-MAML is stochastic gradient descent. However, many other interesting choices exist.

## 3.2   Choices for the Update Operator $U$, and the Exploration Plicy $\pi_\theta$

When only a single iteration of inner policy gradient occurrs, the initial policy $\pi_{\theta_0}$ is entirely responsible for the exploratory sampling. The best exploration would be a policy $\pi_{\theta_0}$ that can generate the most informative samples for identifying the task. However if more than 1 iteration of inner policy gradient occurs, then some of the exploratory behavior can also be attributed to the update operation $\mathcal{U}$. There is a non-trivial interplay between the initial policy and the subsequent policy updates, especially when exploring the environment would take a few different policies.

## 3.3   Understanding the E-MAML Update

A standard reinforcement learning objective can be represented by the stochastic computation graph [29] as in Figure.1a, where the loss is computed w.r.t. the policy parameter $\theta$ using estimator Eq.1. For clarity, we can use a short-hand notation $U$ as in $\theta' = U(\theta)$ to represent this graph (Fig.1b).

**Algorithm 1** E-MAML

**Require:** Task distribution: $P(\mathcal{T})$
**Require:** $\alpha, \beta$ learning step size hyperparameters
**Require:** $n_{\text{inner}}, n_{\text{meta}}$ number of gradient updates for per-task and meta learning
1: **function** $U^k(\varphi, \mathcal{L}, \tau_{[0,\cdots,k-1]})$                                                      ▷ Inner Update Function
2:     **for** i in $[0, \cdots, \text{k-1}]$ **do**
3:         $\varphi \leftarrow \varphi - \alpha \nabla_\varphi \mathcal{L}(\varphi, \tau_i)$
4:     **return** $\varphi$
5: **procedure** E-MAML$(\theta, \phi)$
6:     randomly initialize $\theta$ and $\phi$
7:     **while** not done **do**
8:         Sample batch of tasks $\mathcal{T}_i$ from $p(\mathcal{T})$
9:         **for** $\mathcal{T}_i \in \mathcal{T}$ **do**
10:            Sample rollouts $[\tau]_{[0:n_{\text{inner}}-1]} \sim \pi_\theta$;
11:            $\theta^n \leftarrow U^{n_{\text{inner}}}(\theta, \mathcal{L}, \tau_{[0:n_{\text{inner}}-1]})$                   ▷ High-order update
12:            with $\pi_{\theta^{n_{\text{inner}}}}$, sample $\tau_i^{\text{meta}} = \{s, a, r\}$ from task $\mathcal{T}_i$;
13:         **for** i in $n_{\text{meta}}$ **do**                               ▷ Meta-update the model prior $\theta$
14:            $\theta \leftarrow \theta - \beta \sum_{\tau_i^{\text{meta}}} \nabla_\theta \mathcal{L}_{\text{meta}}[\pi_{U^n_{(\theta, \mathcal{L})}}, \tau_i^{\text{meta}}]$              ▷ no resample

As described in [1], we can then write down the surrogate objective of E-MAML as the graph in Fig.1c whereas model-agnostic meta-learning (MAML) treats the graph as if the first policy gradient update is deterministic. It is important to note that one can in fact smoothly tune between the stochastic treatment of $U$ (Fig.1c) and the non-stochastic treatment of $U$ (Fig.1d) by choosing the hyperparameter $\lambda$. In this regard, E-MAML considers the sampling distribution $\pi_{\theta'}$ as an extra, exploration friendly term that one could add to the standard meta-RL objective used in MAML.

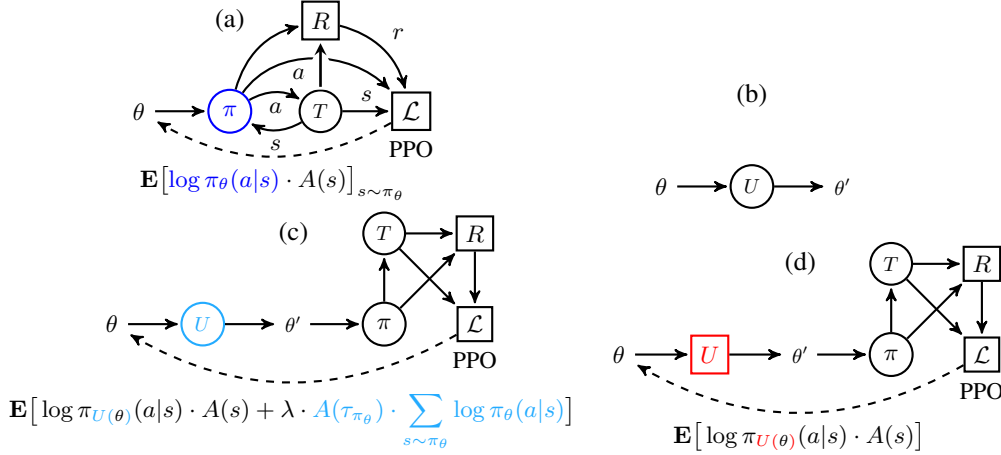

Figure 1: Computation graphs for **(a)** REINFORCE/Policy Gradient, where $\theta$ is the policy parameter. $\pi$ is the policy function. During learning, the policy $\pi_\theta$ interacts with the environment, parameterized as the transition function $T$ and the reward function $R$. $\mathcal{L}$ is the proximal policy optimization objective, which allows making multiple gradient steps with the same trajectory sample. Only 1-meta gradient updated is used in these experiments. **(b)** shows the short-hand we use to represent (a) in (c) and (d). Note that circle $\circ$ represents stochastic nodes, and $\square$ represents deterministic nodes. The policy gradient update sub-graph $U$ is stochastic, which is where here in (b) we have a circle. **(c)** the inner policy gradient sub-graph and the policy gradient meta-loss. The original parameter $\theta$ gets updated to $\theta'$, which is then evaluated by the outer proximal policy optimization objective. E-MAML treats the inner policy gradient update as an stochastic node. Whereas **(d)** MAML treats this as a deterministic node, thus neglecting the causal entropy term of the inner update operator $U$.

### 3.4 E-RL$^2$

RL$^2$ optimizes (1) by feeding multiple rollouts from multiple different MDPs into an RNN. The hope is that the RNN hidden state update $h_t = C(x_t, h_{t-1})$, will learn to act as the update function $U$. Then, performing a policy gradient update on the RNN will correspond to optimizing the meta objective (1).

We can write the RL$^2$ update rule more explicitly in the following way. Suppose $L$ represents an RNN. Let $\text{Env}_k(a)$ be a function that takes an action, uses it to interact with the MDP representing task $k$, and returns the next observation $o$ [3], reward $r$, and a termination flag $d$. Then we have

$$x_t = [o_{-1}, a_{t-1}, r_{t-1}, d_{t-1}] \tag{5}$$
$$[a_t, h_{t+1}] = L(h_t, x_t) \tag{6}$$
$$[o_t, r_t, d_t] = \text{Env}_k(a_t) \tag{7}$$

To train this RNN, we sample $N$ MDPs from $\mathcal{M}$ and obtain $k$ rollouts for each MDP by running the MDP through the RNN as above. We then compute a policy gradient update to move the RNN parameters in a direction which maximizes the returns over the $k$ trials performed for each MDP.

Inspired by our derivation of E-MAML, we attempt to make RL$^2$ better account for the impact of its initial sampling distribution on its final returns. However, we will take a slightly different approach. Call the rollouts that help account for the impact of this initial sampling distribution Explore-rollouts. Call rollouts that do not account for this dependence Exploit-rollouts. For each MDP $M_i$, we will sample $p$ Explore-rollouts and $k - p$ Exploit-rollouts. During an Explore-rollout, the forward pass through the RNN will receive all information. However, during the backwards pass, the rewards contributed during Explore-rollouts will be set to zero. The graident computation will only get rewards provided by Exploit-Rollouts. For example, if there is one Explore-rollout and one Exploit-rollout, then we would proceed as follows. During the forward pass, the RNN will receive all information regarding rewards for both episodes. During the backwards pass, the returns will be computed as

$$R(x_i) = \sum_{j=i}^{T} \gamma^j r_j \cdot \chi_E(x_j) \tag{8}$$

Where $\chi_E$ is an indicator that returns 0 if the episode is an Explore-rollout and 1 if it is an Exploit-rollout. This return, and not the standard sum of discounted returns, is what is used to compute the policy gradient. The hope is that zeroing the return contributions from Explore-rollouts will encourage the RNN to account for the impact of casting a wider sampling distribution on the final meta-reward. That is, during Explore-rollouts the policy will take actions which may not lead to immediate rewards but rather to the RNN hidden weights that perform better system identification. This system identification will in turn lead to higher rewards in later episodes.

## 4 Experiments

### 4.1 Krazy World Environment

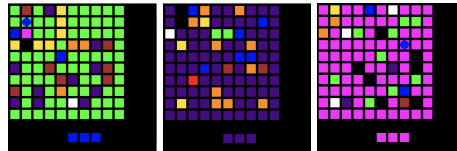

To test the importance of correctly differentiating through the sampling process in meta reinforcement learning, we engineer a new environment known as Krazy World. To succeed at Krazy World, a successful meta learning agent will first need to identify and adapt to many different tile types, color palettes, and dynamics. This environment is challenging even for state-of-the-art RL algorithms. In

Figure 2: Three example worlds drawn from the task distribution. A good agent should first complete a successful system identification before exploiting. For example, in the leftmost grid the agent should identify the following: 1) the orange squares give +1 reward, 2) the blue squares replenish energy, 3) the gold squares block progress, 4) the black square can only be passed by picking up the pink key, 5) the brown squares will kill it, 6) it will slide over the purple squares. The center and right worlds show how these dynamics will change and need to be re-identified every time a new task is sampled.

this environment, it is essential that meta-updates account for the impact of the original sampling distribution on the final meta-updated reward. Without accounting for this impact, the agent will not receive the gradient of the per-task episodes with respect to the meta-update. But this is precisely the gradient that encourages the agent to quickly learn how to correctly identify parts of the environment. See the Appendix for a full description of the environment.

## 4.2 Mazes

A collection of maze environments. The agent is placed at a random square within the maze and must learn to navigate the twists and turns to reach the goal square. A good exploratory agent will spend some time learning the maze's layout in a way that minimizes repetition of future mistakes. The mazes are not rendered, and consequently this task is done with state space only. The mazes are $20 \times 20$ squares large.

## 4.3 Results

In this section, we present the following experimental results.

1. Learning curves on Krazy World and mazes.

2. The gap between the agent's initial performance on new environments and its performance after updating. A good meta learning algorithm will have a large gap after updating. A standard RL algorithm will have virtually no gap after only one update.

3. Three experiments that examine the exploratory properties of our algorithms.

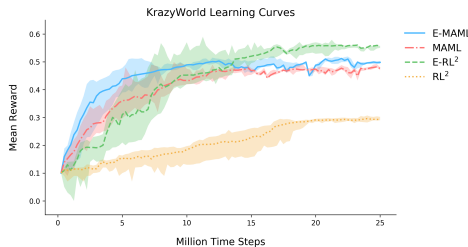

Figure 3: Meta learning curves on Krazy World. We see that E-RL$^2$ achieves the best final results, but has the highest initial variance. Crucially, E-MAML converges faster than MAML, although both algorithms do manage to converge. RL$^2$ has relatively poor performance and high variance. A random agent achieves a score of around 0.05.

When plotting learning curves in Figure 3 and Figure 4, the Y axis is the reward obtained after training at test time on a set of 64 held-out test environments. The X axis is the total number of environmental time-steps the algorithm has used for training. Every time the environment advances forward by one step, this count increments by one. This is done to keep the timescale consistent across meta-learning curves.

For Krazy World, learning curves are presented in Figure 3. E-MAML and E-RL$^2$ have the best final performance. E-MAML has the steepest initial gains for the first 10 million time-steps. Since meta-learning algorithms are often very expensive, having a steep initial ascent is quite valuable. Around 14 million training steps, E-RL$^2$ passes E-MAML for the best performance. By 25 million time-steps, E-RL$^2$ has converged. MAML delivers comparable final performance to E-MAML. However, it takes it much longer to obtain this level of performance. Finally, RL$^2$ has comparatively poor performance on this task and very high variance. When we manually examined the RL$^2$ trajectories to figure out why, we saw the agent consistently finding a single goal square and then refusing to explore any

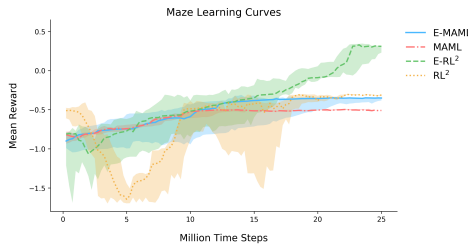

Figure 4: Meta learning curves on mazes. Figure 5 shows each curve in isolation, making it easier to discern their individual characteristics. E-MAML and E-RL$^2$ perform better than their counterparts.

further. The additional experiments presented below seem consistent with this finding.

Learning curves for mazes are presented in Figure 4. Here, the story is different than Krazy World. RL$^2$ and E-RL$^2$ both perform better than MAML and E-MAML. We suspect the reason for this is

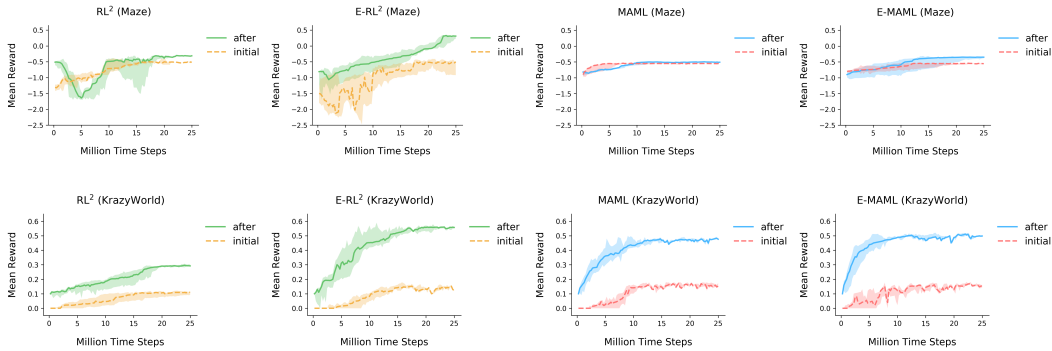

Figure 5: Gap between initial performance and performance after one update. All algorithms show some level of improvement after one update. This suggests meta learning is working, because normal policy gradient methods learn nothing after one update.

that RNNs are able to leverage memory, which is more important in mazes than in Krazy World. This environment carries a penalty for hitting the wall, which MAML and E-MAML discover quickly. However, it takes E-RL$^2$ and RL$^2$ much longer to discover this penalty, resulting in worse performance at the beginning of training. MAML delivers worse final performance and typically only learns how to avoid hitting the wall. RL$^2$ and E-MAML sporadically solve mazes. E-RL$^2$ manages to solve many of the mazes.

When examining meta learning algorithms, one important metric is the update size after one learning episode. Our meta learning algorithms should have a large gap between their initial policy, which is largely exploratory, and their updated policy, which should often solve the problem entirely. For MAML, we look at the gap between the initial policy and the policy after one policy gradient step For RL$^2$, we look at the results after three exploratory episodes, which give the RNN hidden state $h$ sufficient time to update. Note that three is the number of exploratory episodes we used during training as well. This metric shares similarities with the Jump Start metric considered in prior literature [35]. These gaps are presented in figure 5.

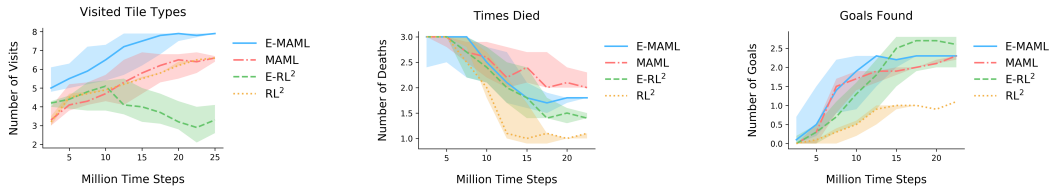

Figure 6: Three heuristic metrics designed to measure an algorithm's system identification ability on Krazy World: Fraction of tile types visited during test time, number of times killed at a death square during test time, and number of goal squares visited. We see that E-MAML is consistently the most diligent algorithm at checking every tile type during test time. Improving performance on these metrics indicates the meta learners are learning how to do at least some system identification.

Finally, in Figure 6 we see three heuristic metrics desigend to measure a meta-learners capacity for system identification. First, we consider the fraction of tile types visited by the agent at test time. A good agent should visit and identify many different tile types. Second, we consider the number of times an agent visits a death tile at test time. Agents that are efficient at identification should visit this tile type exactly once and then avoid it. More naive agents will run into these tiles repeatedly, dying repetedly and instilling a sense of pity in onlookers. Finally, we look at how many goals the agent reaches. RL$^2$ tends to visit fewer goals. Usually, it finds one goal and exploits it. Overall, our suggested algorithms achieve better performance under these metrics.

## 5   Related Work

This work builds depends upon recent advances in deep reinforcement learning. [15, 16, 13] allow for discrete control in complex environments directly from raw images. [28, 16, 27, 14], have allowed for high-dimensional continuous control in complex environments from raw state information.

It has been suggested that our algorithm is related to the exploration vs. exploitation dilemma. There exists a large body of RL work addressing this problem [10, 5, 11]. In practice, these methods are often not used due to difficulties with high-dimensional observations, difficulty in implementation on arbitrary domains, and lack of promising results. This resulted in most deep RL work utilizing epsilon greedy exploration [15], or perhaps a simple scheme like Boltzmann exploration [6]. As a result of these shortcomings, a variety of new approaches to exploration in deep RL have recently been suggested [34, 9, 31, 18, 4, 19, 17, 8, 23, 17, 8, 26, 25, 32, 33, 12, 24]. In spite of these numerous efforts, the problem of exploration in RL remains difficult.

Many of the problems in meta RL can alternatively be addressed with the field of *hierarchical reinforcement learning*. In hierarchical RL, a major focus is on learning primitives that can be reused and strung together. Frequently, these primitives will relate to better coverage over state visitation frequencies. Recent work in this direction includes [38, 2, 36, 20, 3, 39]. The primary reason we consider meta-learning over hierarchical RL is that we find hierarchical RL tends to focus more on defining specific architectures that should lead to hierarchical behavior, whereas meta learning instead attempts to directly optimize for these behaviors.

As for meta RL itself, the literature is spread out and goes by many different names. There exist relevant literature on life-long learning, learning to learn, continual learning, and multi-task learning [22, 21]. We encourage the reviewer to look at the review articles [30, 35, 37] and their citations. The work most similar to ours has focused on adding curiosity or on a free learning phrase during training. However, these approaches are still quite different because they focus on defining an intrinsic motivation signals. We only consider better utilization of the existing reward signal. Our algorithm makes an explicit connection between free learning phases and the its affect on meta-updates.

## 6   Closing Remarks

In this paper, we considered the importance of sampling in meta reinforcement learning. Two new algorithms were derived and their properties were analyzed. We showed that these algorithms tend to learn more quickly and cover more of their environment's states during learning than existing algorithms. It is likely that future work in this area will focus on meta-learning a curiosity signal which is robust and transfers across tasks, or learning an explicit exploration policy. Another interesting avenue for future work is learning intrinsic rewards that communicate long-horizon goals, thus better justifying exploratory behavior. Perhaps this will enable meta agents which truly *want* to explore rather than being forced to explore by mathematical trickery in their objectives.

## 7   Acknowledgement

This work was supported in part by ONR PECASE N000141612723 and by AWS and GCE compute credits.

## Appendix A: Krazy-World

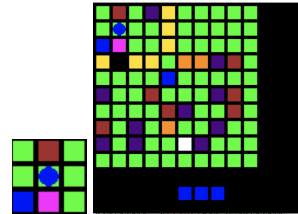

We find this environment challenging for even state-of-the-art RL algorithms. For each environment in the test set, we optimize for 5 million steps using the Q-Learning algorithm from Open-AI baselines. This exercise delivered a mean score of 1.2 per environment, well below the human baselines score of 2.7. The environment has the following challenging features:

Figure 7: A comparison of local and global observations for the Krazy World environment. In the local mode, the agent only views a $3 \times 3$ grid centered about itself. In global mode, the agent views the entire environment.

**8 different tile types**: *Goal squares* provide +1 reward when retrieved. The agent reaching the goal does not cause the episode to terminate, and there can be multiple goals. *Ice squares* will be skipped over in the direction the agent is transversing. *Death squares* will kill the agent and end the episode. *Wall squares* act as a wall, impeding the agent's movement. *Lock squares* can only be passed once the agent has collected the corresponding key from a key square. *Teleporter squares* transport the agent to a different teleporter square on the map. *Energy squares* provide the agent with additional energy. If the agent runs out of energy, it can no longer move. The agent proceeds normally across *normal squares*.

**Ability to randomly swap color palette**: The color palette for the grid can be permuted randomly, changing the color that corresponds to each of the tile types. The agent will thus have to identify the new system to achieve a high score. Note that in representations of the gird wherein basis vectors are used rather than images to describe the state space, each basis vector corresponds to a tile type–permuting the colors corresponds to permuting the types of tiles these basis vectors represent. We prefer to use the basis vector representation in our experiments, as it is more sample efficient.

**Ability to randomly swap dynamics**: The game's dynamics can be altered. The most naive alteration simply permutes the player's inputs and corresponding actions (issuing the command for down moves the player up etc). More complex dynamics alterations allow the agent to move multiple steps at a time in arbitrary directions, making the movement more akin to that of chess pieces.

**Local or Global Observations**: The agent's observation space can be set to some fixed number of squares around the agent, the squares in front of the agent, or the entire grid. Observations can be given as images or as a grid of basis vectors. For the case of basis vectors, each element of the grid is embedded as a basis vector that corresponds to that tile type. These embeddings are then concatenated together to form the observation proper. We will use local observations.

## Appendix B: Table of Hyperparameters

Table 1: Table of Hyperparameters: E-MAML

| Hyperparameter | Value | Comments |
|---|---|---|
| Parallel Samplers | $40 \sim 128$ | |
| Batch Timesteps | 500 | |
| Inner Algo | PPO / VPG / CPI | Much simpler than TRPO |
| Meta Algo | PPO / VPG / CPI | |
| Max Grad Norm | $0.9 \sim 1.0$ | improves stability |
| PPO Clip Range | 0.2 | |
| Gamma | $0 \sim 1$ | tunnable hyper parameter |
| GAE Lambda | 0.997 | |
| Alpha | 0.01 | Could be learned |
| Beta | 1e-3 | 60 |
| Vf Coeff | 0 | value baseline is disabled |
| Ent Coeff | 1e-3 | improves training stability |
| Inner Optimizer | SGD | |
| Meta Optimizer | Adam | |
| Inner Gradient Steps | $1 \sim 20$ | |
| Simple Sampling | True/False | With PPO being the inner algorithm, one can reuse the same path sample for multiple gradient steps. |
| Meta Gradient Steps | $1 \sim 20$ | Requires PPO when $> 1$ |

## Appendix C: Experiment Details

For both Krazy World and mazes, training proceeds in the following way. First, we initialize 32 training environments and 64 test environments. Every initialized environment has a different seed. Next, we initialize our chosen meta-RL algorithm by uniformly sampling hyper-parameters from predefined ranges. Data is then collected from all 32 training environments. The meta-RL algorithm then uses this data to make a meta-update, as detailed in the algorithm section of this paper. The meta-RL algorithm is then allowed to do one training step on the 64 test environments to see how fast it can train at test time. These test environment results are recorded, 32 new tasks are sampled from the training environments, and data collection begins again. For MAML and E-MAML, training at test time means performing one VPG update at test time (see figure **??** for evidence that taking only one gradient step is sufficient). For $RL^2$ and $E\text{-}RL^2$, this means running three exploratory episodes to allow the RNN memory weights time to update and then reporting the loss on the fourth and fifth episodes. For both algorithms, meta-updates are calculated with PPO [27]. The entire process from the above paragraph is repeated from the beginning 64 times and the results are averaged to provide the final learning curves featured in this paper.

## Footnotes

[2]Half-cheetah is an example of a weak benchmark–it can be learned in just a few gradient step from a random prior. It can also be solved with a linear policy. Stop using half-cheetah.

[3]RL$^2$ works well with POMDP's because the RNN is good at system-identification. This is the reason why we chose to use $o$ as in "observation" instead of $s$ for "state" in this formulation.

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
