[Reviews · NeurIPS 2018]

Reviewer 1



The paper shows the importance of the used training setup for MAML and RL^2. A setup can include "exploratory episodes" and measure the loss only on the next "reporting" episodes. The paper presents interesting results. The introduced E-MAML and E-RL^2 variants clearly help. The main problem with the paper: The paper does not define well the objective. I only deduced from the Appendix C that the setup is: After starting in a new environment, do 3 exploratory episodes and report the collected reward on the next 2 episodes. This is not explained by the Equation (1). The paper would be much clearer if mentioning the exploratory episodes. The paper has a potential to help people to design the right training objective. Comments to improve the paper: 1) Mention early the dependency of the update operator on the exploratory episodes. 2) If you explicitly define the objective, the RL^2 on the right objective will become the same as E-RL^2. 3) You should discuss the important design choices, when designing the objective or its approximation: 3.1) The small number of "exploratory" and "reporting" episodes may be just an approximation for the large number of episodes seen during training and testing. Some high-hanging exploration would appear beneficial only if having many reporting episodes. 3.2) The reporting phase should start with a new episode to prevent the agent placing a bag of gold on the ground before the end of the exploratory phase. 3.3) The length of the exploratory phase should not affect the discount used during the reporting phase. Otherwise the agent would like to end the exploratory phase early. Reconsider Equation (11). 3.4) The length of the exploratory phase should be limited to avoid exploring forever. Update: I have read the rebuttal. Thank you for considering the clarification improvements. I hope that you will specify the loss function as a function of the exploratory and the reporting episodes. Only then it makes sense to show that the other approaches are not minimizing the loss function directly.

Reviewer 2



This paper proposes extensions to two popular meta-learning techniques: MAML and RL^2, with the aim of learning an optimal sampling strategy (policy) to maximize expected returns that takes into account the effect of the initial sampling policy on final return. This was more theoretically derived for MAML by explicitly including the dependency in the expectation and differentiating to compute the appropriate gradient, which turns out to have an additional term compared to MAML. This additional term allows the policy to be more exploratory. RL^2 was modified similarly by including multiple rollouts per update, p of which contribute 0 return to the backwards pass (exploratory episodes) and k-p of which contribute the normal return. These models were evaluated on a new environment “Krazy World”, which has dynamically generated features every episode, and also a maze environment. The authors propose a simple but useful idea, and I think it does make a contribution. Although there are a lot of interesting ideas and results, it feels like a work in progress rather than a complete piece of work. Some technical details are missing, which impact the clarity and quality, as I’ve listed below. This makes me concerned about the reproducibility of their results. The new environment Krazy World is an interesting benchmark task for meta-learning, and I’d like to see it open-sourced if possible. More specific comments: -It’s interesting that E-MAML does better initially but E-RL^2 does better in the end. What level of reward is optimal for the two tasks? -For E-RL^2, have you explored tuning the relative contributions of exploration on final return? That is, how is performance impacted by performing more exploratory rollouts per update? How did you settle on 3 and 2? -Are the maze layouts procedurally generated every episode? -There are a lot of typos, please proofread more carefully (e.g. page 7 “desigend” -> “designed”, page 8 “defining an intrinsic motivation signals” -> “signal” and “and the its affect” -> “and its effect”) -In section 2.5, k is both the number of rollouts for each MDP and also the task the MDP represents, so this is confusing -It’s very strange that they don’t cite Duan et al’s RL^2 paper at all, since E-RL^2 is named after it -Figure 8 (in the appendix) is not helpful at all, the axes and legends are impossible to read and axes aren’t even labeled. Why are there 3 columns? The caption only mentions 2. -What do the error shades represent in the plots? Standard deviation? Standard error? Over how many runs? Update after reading the rebuttal: It seems like all the reviewers had major concerns regarding clarity, and the authors' response does seem to address many of these. There are a lot of things to revise however, and I'm not completely convinced the final product will be ready for publication at NIPS, given how in-progress the initial version felt. Given this, I can't at this time change my assessment, although I still tentatively recommend acceptance and encourage the authors to revise with clarity in mind.

Reviewer 3



i had some difficulty in understanding what was being done. but this paper is outside of my expertise, so it may be mostly my fault. let me start by checking whether i can correctly summarize the two methods (one with gradient descent as the inner learning algorithm, and one with an RNN as the inner learning algorithm). in standard MAML, you might do a behavioral trajectory, then do a gradient step using the rewards and actions sampled in that trajectory, and then do another trajectory. the outer learning algorithm optimizes the initial parameters such that the policy after the gradient step leads to higher rewards. E-MAML adds another term to be optimized, which encourages you to do actions *before* the gradient step which lead to a good policy *after* the gradient step. in other words, it encourages you to do actions that lead to the most useful gradients. in standard RL^2, you use policy gradient as the outer learner to train an RNN (which receives rewards as inputs) to be a good inner learner. if i understood correctly, the only modification in E-RL^2 is to ignore rewards (for the sake of policy gradients, but still providing them as inputs to the RNN) during a randomly chosen subset of rollouts. this is meant to avoid trapping the learning process in exploiting immediately available rewards. the paper also introduces a task environment where some of the rules are frequently re-randomized so that exploration is important. the results are currently minimal; it's a little bit difficult to know what to take away from them. i would be interested to see more analysis of the performance, including robustness to hyperparameters for each algorithm. it would also be nice to see more analysis of what effect the extra optimization objective in E-MAML has on the policy, versus MAML. is it possible to show more detail of what the exploration looks like in krazyworld or in the mazes? stray parenthesis in equation (1) tau isn't defined when it's introduced in equation (2) missing a marker for which is equation (3)